# On topology and knotty entanglement in protein folding

**Alexander Begun[1], Sergei Liubimov[1], Alexander Molochkov[1], Antti J. Niemi[1,2,3]\***

**1** Pacific Quantum Center, Far Eastern Federal University, Vladivostok, Russia, **2** Nordita, Stockholm University, Stockholm, Sweden, **3** Department of Physics, Beijing Institute of Technology, Haidian District, Beijing, People's Republic of China

\* Antti.Niemi@su.se

**Data Availability Statement:** All relevant data are within the manuscript and its Supporting information files.

**Funding:** This work was supported by the Ministry of Science and Higher Education of Russia in the form of a grant awarded to AM (0657-2020-0015),

## Abstract

We investigate aspects of topology in protein folding. For this we numerically simulate the temperature driven folding and unfolding of the slipknotted archaeal virus protein AFV3-109. Due to knottiness the (un)folding is a topological process, it engages the entire backbone in a collective fashion. Accordingly we introduce a topological approach to model the process. Our simulations reveal that the (un)folding of AFV3-109 slipknot proceeds through a folding intermediate that has the topology of a trefoil knot. We observe that the final slipknot causes a slight swelling of the folded AFV3-109 structure. We disclose the relative stability of the strands and helices during both the folding and unfolding processes. We confirm results from previous studies that pointed out that it can be very demanding to simulate the formation of knotty self-entanglement, and we explain how the problems are circumvented: The slipknotted AFV3-109 protein is a very slow folder with a topologically demanding pathway, which needs to be properly accounted for in a simulation description. When we either increase the relative stiffness of bending, or when we decrease the speed of ambient cooling, the rate of slipknot formation rapidly increases.

## Introduction

Topological techniques are often very powerful. They are used with great success to analyze numerous problems in Physics, from theories of fundamental interactions to models of condensed matter. Here we propose to introduce topological techniques to study protein folding and dynamics, where thus far these techniques have been used only sparsely. As an example we analyze the formation of knots and self-entanglement in protein folding. A knot along a protein backbone is a delocalized structure, with a definite topological character. A knot can not be removed by any small amplitude local motion such as twisting, bending or crumpling of the protein backbone. A knotty self-entanglement persists as long as there is no backbone chain crossing, and provided the C and N terminals of the protein are not circumnavigated.

For a long time it was thought that there are no, or only very few knotted proteins [1–3]. But subsequently it was found and concluded, that proteins with a knotty self-entanglement are quite common [3–12] among crystallographic Protein Data Bank (PDB) [13] structures. Dedicated servers and databases are now being developed to identify, analyze and classify

the Swedish Research Council in the form of a contract awarded to AJN (2018-04411), and the Carl Trygger Foundation in the form of a grant awarded to AJN (CTS 18: 276). AJN acknowledges the support of a Qian Ren grant at BIT. The research by AJN is also part of a COST Action collaboration (CA17139).

**Competing interests:** The authors have declared that no competing interests exist.

knotty protein structures [14–16]. It has been estimated that as many as six per cent of all globular protein structures display some level of self-entangled complexity [17–19].

Topology is a global, collective property of a system. Thus, in the analysis of topological structures, such as knotty entanglements in a protein, global techniques that are based on topological concepts should be preferable. However, the prevailing ambition in the protein modeling community is to simulate the entire folding process as atomistically as possible, within the limitations of available computer resources. The commonly available techniques aim to describe protein folding primarily as a local process, with little if any regard to global aspects such as the backbone topology. Instead, global aspects of protein structure and dynamics are presumed to only reflect the local character of native contacts, and other local details. For example, in all-atom simulations the interaction range between individual atoms is commonly cut off beyond 10-12 Å, and in Gō-type models the range of interactions can be even shorter. Since knottiness is not a local, but a global characteristic of a protein backbone, we suspect that such a preference to localize all interactions is a reason why it appears to be very hard to simulate the folding of a knotty protein with a high success rate. At least unless one introduces some kind of bias such as funneling, steering or other method of (global) augmentation [5, 6, 11, 12, 19], to cross the topological barrier.

Here we introduce and develop global techniques, as a complement to the already existing local ones, to account for topological aspects of protein folding dynamics. For this we scrutinize the folding and unfolding processes of a protein with a knotty self-entanglement. Our approach builds on an effective theory description. We employ a mechanical free energy that describes the entire protein tertiary structure as a single topological object, in lieu of the individual atoms or other highly localized interaction centers. The protein is modeled as a topological multi-soliton solution of a nonlinear difference equation that determines the critical points of a mechanical free energy; the equation that identifies the critical points resembles the discrete nonlinear Schrödinger (DNLS) equation [20, 21].

Soliton solutions of non-linear difference and differential equations are commonly encountered when searching for principles of structural self-organization in physical scenarios. The nonlinear Schrödinger equation is the paradigm equation for describing topological solitons. In our approach an individual topological soliton models a single super-secondary protein structure such as a helix-loop-helix or a strand-loop-strand motif. We use the DNLS equation to combine together several mutually interacting individual soliton profiles into a multi-soliton that models the entire $C\alpha$ backbone as a single entity. From the point of view of an energy landscape description [22], the multi-soliton is a stable attractor in the landscape of all conceivable protein structures. The mechanical free energy of the DNLS equation funnels unfolded protein structures to progress towards its minimum energy configuration, described by the multi-soliton.

Our multi-soliton description is computationally highly effective: When we start from a random chain, the folding simulation proceeds several orders of magnitude faster than in any other computational approach to protein folding that we are aware of. Thus we can perform numerical simulations to analyze the mechanism of protein self-entanglement and the dynamics of knot formation, with a very high level of computational efficiency using a simple laptop computer.

By its design, our mechanical free energy with its multi-soliton critical points models the protein backbone at very low temperature, with no consideration to thermal fluctuations. In order to study the folding and unfolding patterns, we subject the multi-soliton to a series of repeated heating and cooling simulations. For this we use Monte Carlo methodology and the Glauber algorithm that we adapt to proteins [23], to describe their non-equilibrium thermodynamics in the presence of variable ambient simulation temperature. In our simulation we

follow how a knotted multi-soliton unfolds into a random chain when the ambient temperature increases. We then continue and follow how the random chain folds and self-entangles back into the knotty multi-soliton configuration, as the ambient temperature decreases. By varying the rate of temperature change we identify and investigate the different folding and unfolding patterns and pathways.

We start and construct a multi-soliton that models the three-dimensional C$\alpha$ backbone of the folded AFV3-109 protein as a solution of the generalized DNLS equation. We use the crystallographic Protein Data Bank (PDB) structure with PDB code 2J6B [24] as a decoy. The AFV3-109 is an $\alpha/\beta$ protein that, when folded, supports a self-entangled structure akin the shoelace slipknot. Even though the slipknot is technically not a proper knot, it has nevertheless a topological character, with a large degree of structural stability under local backbone deformations; the topological character of the slipknotted AFV3-109 is similar to that encountered in other, more elaborately self-entangled knotty proteins [6, 17–19]. Moreover, our simulation results reveal that in the case of the AFV3-109 protein, the formation of the slipknot entails a trefoil knot as a folding intermediate. The AFV3-109 protein has been studied previously as a prototype example of a slipknotted protein [6, 24], but these previous simulations have not reported of an intermediate trefoil structure. On the contrary, it is presumed that a slipknot itself occurs as an intermediate, along the folding pathway of more complex knot formation, including a trefoil [6, 17–19].

## Methods

We model protein structures in terms of *topological solitons* [20, 21]. These are extended objects that emerge as a non-local solution to a set of partial difference equations, and are topologically stable against decay to a "trivial" solution. The pertinent partial difference equations describe critical points of an energy function, in an effective theory approach that builds on the (virtual) C$\alpha$ protein backbone geometry. For this we use the formalism of discrete Frenet framing [25]: At the location of the $i^{th}$ C$\alpha$ atom, a discrete Frenet frame comprises the mutually orthonormal backbone tangent ($\mathbf{t}_i$), binormal ($\mathbf{b}_i$) and normal ($\mathbf{n}_i$) vectors, as follows

$$\mathbf{t}_i = \frac{\mathbf{r}_{i+1} - \mathbf{r}_i}{|\mathbf{r}_{i+1} - \mathbf{r}_i|} \qquad \& \qquad \mathbf{b}_i = \frac{\mathbf{t}_{i-1} \times \mathbf{t}_i}{|\mathbf{t}_{i-1} \times \mathbf{t}_i|} \qquad \& \qquad \mathbf{n}_i = \mathbf{b}_i \times \mathbf{t}_i \tag{1}$$

where $\mathbf{r}_i$ ($i = 1, \ldots, n$) are the C$\alpha$ coordinates. These vectors are subject to the discrete Frenet equation [25]

$$\begin{pmatrix} \mathbf{n}_{i+1} \\ \mathbf{b}_{i+1} \\ \mathbf{t}_{i+1} \end{pmatrix} = \exp\{-\theta_i T^2\} \exp\{-\phi_i T^3\} \begin{pmatrix} \mathbf{n}_i \\ \mathbf{b}_i \\ \mathbf{t}_i \end{pmatrix} \tag{2}$$

Here the $\theta$ are the virtual backbone bond angles, they are computed as follows,

$$\theta_i = \arccos(\mathbf{t}_{i+1} \cdot \mathbf{t}_i) \tag{3}$$

The virtual backbone torsion angles $\phi$ are computed as follows,

$$\phi_i = \text{sign}[(\mathbf{b}_{i-1} \times \mathbf{b}_i) \cdot \mathbf{t}_i] \cdot \arccos(\mathbf{b}_{i+1} \cdot \mathbf{b}_i) \tag{4}$$

The $T^2$ and $T^3$ are $3 \times 3$ matrices that generate three dimensional rotations with $(T^i)_{jk} = \epsilon_{ijk}$, the permutation symbol. By substituting (1) in (3), (4) we can compute $(\theta_i, \phi_i)$ in terms of the

C$\alpha$ coordinates $\mathbf{r}_i$. Conversely, when we know $(\theta_i, \phi_i)$ and the distances between neighboring C$\alpha$ atoms, we can reconstruct the C$\alpha$ coordinates $\mathbf{r}_i$ by solving (2). Here we assume that the distances between neighboring C$\alpha$ atoms coincides with the average PDB value

$$|\mathbf{r}_{i+1} - \mathbf{r}_i| \sim 3.8\,\text{Å}$$

Once the C$\alpha$ position have been determined, the remaining heavy atom structure can be fully reconstructed from the knowledge of the $(\theta, \phi)$ coordinates [26–31], even in the case of a dynamical protein [32]. Thus, in the sequel we do not explicitly include the effects of side chains.

The set of all possible $(\theta, \phi_i)$ values governs the entire conformational space of the C$\alpha$ backbone. Various effective theory conformational free energy functions have been previously constructed in terms of these coordinates. Examples include the fully flexible chain model and its extensions [33–35], that are used widely in studies of biological macromolecules and other filamental objects. Thus, we also employ the C$\alpha$ backbone angles $(\theta_i, \phi_i)$ of each and every C$\alpha$ atom, as the dynamical variables to introduce a refined extension of these earlier models. Ours is the following mechanical free energy [23, 36–39]

$$F = \sum_{k \in solitons} \left\{ \sum_{i=1}^{n} \left( -2\theta_{i+1}\theta_i + 2\theta_i^2 + \lambda_k \left(\theta_i^2 - m_k^2\right)^2 + \frac{d_k}{2}\,\theta_i^2\phi_i^2 \right) + \right.$$

$$\left. + \sum_{i=1}^{n} \left( \frac{c_k}{2}\,\phi_i^2 - b_k\,\theta_i^2\phi_i - a_k\,\phi_i \right) \right\} + \sum_{i>j}^{n} V(\mathbf{r}_i - \mathbf{r}_j) \tag{5}$$

We refer to S1 File for additional discussion including background and motivation in terms of integrable models and soliton theory. For the present purposes the following is sufficient:

The mechanical free energy (5) is designed to model the geometry of the natively folded C$\alpha$ backbone as its minimal energy critical point:

- The first sum over index $i$ (that labels the $n$ residues) coincides with the energy function of the discretized non-linear Schrödinger (DNLS) equation in the standard Hasimoto representation [20].

- The second sum over the index $i$ extends the DNLS energy function in a manner appropriate for modeling a folded protein: The first term in this sum, with parameter $c_k$, is a Proca mass term. In combination with the second term of the first sum, it constitutes the Kirchhoff energy of an elastic rod [34]. The second term, with parameter $b_k$, is the conserved momentum in the integrable DNLS hierarchy. The third term, with parameter $a_k$, is the conserved helicity in the DNLS hierarchy. These two terms break the parity symmetry, and make the backbone right-handed chiral (for positive $a_k$, $b_k$).

- The last term in the free energy, a sum over two-body interactions $V(\mathbf{r}_i - \mathbf{r}_j)$, models long distance interactions along the chain. However, a topological soliton is also highly non-local, it is an extended topological object. Thus the topological multi-soliton that we construct to model the C$\alpha$ chain also describes non-local effects due to long range interactions. Thus, to avoid double counting we include in this last term only a hard-core Pauli repulsion that prevents the chain from self-crossing. For this we use a step-wise profile that keeps any two C$\alpha$ atoms at least 3.8 Å apart. We refer to [23] for detailed analysis of different choices $V(\mathbf{r}_i - \mathbf{r}_j)$.

- Finally, the sum with index $k \in$ solitons extends over all the individual super-secondary loop segments, such as a helix-loop-helix or a strand-loop-strand motif, that characterize the local

C$\alpha$ geometry. The parameters ($\lambda_k$, $m_k$, $a_k$, $b_k$, $c_k$, $d_k$) have constant values in any individual motif, but these parameter values are in general different for different motifs, with different amino acid structures.

Note that all the contributions in (5) are functionals of the various tangent vectors $\mathbf{t}_i$: For the ($\theta_i$, $\phi_i$) dependent terms this follows from (1), (3) and (4) and for the last term we use

$$\mathbf{r}_i - \mathbf{r}_j = \mathbf{t}_{i-1} + \cdots + \mathbf{t}_j \qquad (i > j)$$

Accordingly, the free energy (5) can be interpreted as a leading order contribution in a systematic expansion of a full, complete free energy that depends on the various two-body interactions, through the distances $\mathbf{r}_i - \mathbf{r}_j$ between all point-like interaction centers, expressed in terms of the tangent vectors $\mathbf{t}_i$.

To determine the various parameter values, we demand that the minimum energy critical point of (5) describes the natively folded C$\alpha$ geometry, with a prescribed precision. The critical points of (5) are topological soliton solutions of the following generalized DNLS equation [40, 41],

$$\frac{\delta F}{\delta \theta_i} = 2(2\theta_i - \theta_{i+1} - \theta_{i-1}) + 4\lambda_k(\theta_i^2 - m_k^2)\theta_i + (d_k\phi_i^2 - 2b_k\phi_i)\theta_i = 0 \qquad (6)$$

$$\frac{\delta F}{\delta \phi_i} = (d_k\theta_i^2 + c_k)\phi_i - b_k\theta_i^2 - a_k = 0 \quad \Rightarrow \quad \phi_i = \frac{b_k\theta_i^2 + a_k}{d_k\theta_i^2 + c_k} \qquad (7)$$

An individual topological soliton models a segment in a super-secondary structure such as a helix-loop-helix or a strand-loop-strand motif with constant i.e. fixed-$k$ parameters. For example, a right-handed $\alpha$-helix has

$$\alpha - \text{helix} : \quad \begin{cases} \theta \approx \dfrac{\pi}{2} \\ \phi \approx 1 \end{cases}$$

and for the $\beta$-strand

$$\beta - \text{strand} : \quad \begin{cases} \theta \approx 1 \\ \phi \approx \pi \end{cases}$$

A single soliton is characteristically a loop structure that interpolates between different regular structures such as $\alpha$-helices and $\beta$-strands. But a long loop can also accommodate more than one single soliton. Along an individual single soliton structure the values of ($\theta_i$, $\phi_i$) are variable while the parameters $\lambda_k$, $m_k$, $a_k$, ... have constant values, over the entire single soliton. The multi-soliton solution of (6, 7) combines the individual solitons into a single structure that models the entire protein tertiary structure, with different sets of $k$-dependent parameters that correspond to the different individual soliton segments. The number of individual soliton profiles that are introduced, depend on the desired precision of the multi-soliton: An increase in the number of individual solitons increases the number of parameters, and improves the precision. Since an individual soliton generically extends over several amino acids, the number of parameters in the multi-soliton is usually comparable to, in fact often even smaller than, the number of amino acids in the protein that it describes. A multi-soliton profile that describes a crystallographic protein structure with its experimental resolution, has usually far fewer parameters, and accordingly also much more predictive power, than e.g. Gō-like structure

based models. In particular, the free energy function (5) does not engage any network of native contacts between individual atoms or other localized interaction centers.

We aim to describe (un)folding experiments where the protein folding and unfolding processes are controlled by temperature variations, rather than by e.g. *pH* variations, or denaturants such as inorganic salts and organic solvents. This also sets our approach apart from Gō models [5, 6] that fold the protein at a constant temperature.

There are many techniques to simulate temperature variations, and here we use a Markovian Monte Carlo algorithm. For the Monte Carlo update there are different alternatives, and we refer to [23] for a comprehensive algorithm comparison in the present model. We adopt the following variant: At each Monte Carlo step $n$ we independently change either the torsion angle $\phi_i$ or the bond angle $\theta_i$ at a randomly chosen site $i$ and compute the ensuing change $\Delta F$ in the free energy (5)

$$\Delta F = F(\theta^{n+1}, \phi^{n+1}) - F(\theta^n, \phi^n)$$

We then use a version of the Glauber algorithm [23] to determine the transition probability $P$ between the two states

$$P = \frac{e^{-\beta\Delta F}}{1 + e^{-\beta\Delta F}} \tag{8}$$

In our simulations we adiabatically decrease and increase the value of the temperature factor $\beta$, to simulate heating and cooling. Note that the inverse Monte Carlo temperature factor $\beta^{-1}$ can not be directly identified with the physical temperature factor $k_B T$ where $k_B$ is the Boltzmann constant and $T$ is the ambient temperature measured in Kelvin. But for an equilibrium distribution the two can be related by methods of renormalization, as described in [42].

When the number of updates increases and the value of $\beta$ is kept fixed, the Glauber algorithm is known to approach the Gibbsian equilibrium distribution at an exponential rate. Moreover, the Glauber algorithm models pure relaxation dynamics that for simple systems reproduces Arrhenius' law. At the same time, small proteins are known to fold according to Arrhenius' law [47].

In the case of the torsion angles $\phi_i$ the change in their values at each Monte Carlo step is calculated according to

$$\phi_i^{n+1} \quad = \quad \phi_i^n + \frac{\pi}{2} R \tag{9}$$

where $R$ is a random number with normal distribution centered at $R = 0$ and with dispersion $\Delta R = 1$. The backbone bond angles $\theta_i$ are known to be much stiffer than the torsion angles $\phi_i$. For this reason we calculate the change in their values using the heat bath algorithm described in [23]. Accordingly $\theta_i^{n+1}$ is determined in terms of $\theta^n$ using a random number with probability distribution

$$P_\theta = \frac{1}{Z_\theta} e^{-\beta_{hb} F_\theta} \tag{10}$$

in the interval $[0, \pi)$ where $F_\theta$ is computed from all the $\theta$-dependent terms in (5). We relate the parameter $\beta_{hb}$ to the inverse Monte Carlo temperature factor $\beta$ and in our production simulations we use $\beta_{hb} = 10^{13}\beta$.

The simulation algorithm consists of three stages; heating, thermalization and cooling. After each step we change the value of $\beta$ in (8), by multiplying it with a constant, and the value of the constant determines the rate of heating/cooling. During the high temperature

thermalization stage the number of Monte Carlo steps is always fixed to $7{\times}10^6$ steps and in this stage the value of $\beta$ remains constant. During heating and cooling the number of steps can be variable and in the simulation described here we have $480{\times}10^6$ steps.

A full simulation ensemble consists of 1.000 independent heating and cooling cycles at the given temperature factor value. At the end of each simulation cycle we screen the final conformation for slipknotted structure using a variety of distance measures. These include the overall RMS distance between the final simulated structure and its crystallographic target, in combination with distances between individual key residues. In borderline cases we resort to visual inspection of the final structure, in order to detect/confirm the presence (or absence) of a knotted configuration.

In the case of the archaeal virus protein AFV3-109 studied here, the Protein Data Bank structure 2J6B that we use as a decoy for constructing the multi-soliton solution has 109 amino acids. The core of this $\alpha/\beta$ protein consists of a $\beta$-sheet with five $\beta$-strands that are connected by helices and loops. The residues 78-106 make up a looped segment that is threaded into a slipknot through a knotting loop that consists of residues 8-77: The Fig 1 shows the slipknot structure.

For identifying the soliton profiles we use the software *GaugeIT* and for determining the parameter values we use the software *Propro*. Both are accessible through https://protoin.ru/propro/index.php.

## Results

We have constructed the multi-soliton that describes the C$\alpha$ backbone of 2J6B as a minimum energy critical point of the mechanical free energy (5). According to [24] there are 3 helical and 5 strand-like regular segments in the crystallographic structure. Thus, including the C and N terminals, there are at least 9 individual loops along the relatively short backbone This suggests us to start and introduce a division of the C$\alpha$ backbone into 9 individual solitons. However, a scrutiny of the crystallographic 2J6B structure reveals that it has a more complex geometry. There are relatively long loops between residues 6-18, 57-74 and 81-91, and some of the $\beta$-strands are slightly bent. In order to describe the backbone with a precision that is

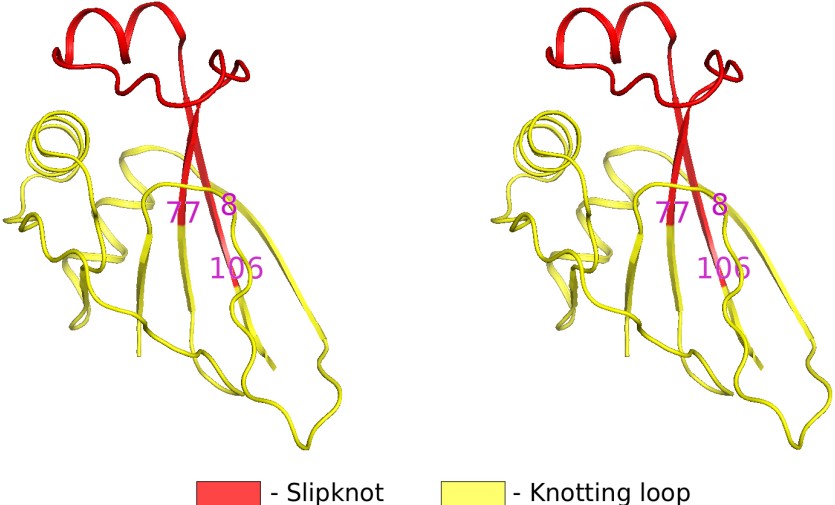

**- Slipknot**     **- Knotting loop**

**Fig 1. A cross-eye 3D view of the AFV3-109 (PDB code 2J6B), the yellow knotting loop extends over residues 8-77 and the red threaded slipknot consists of residues 78-106.**

comparable to the resolution of the crystallographic structure we then introduce additional soliton structures: We identify a total of 20 individual soliton profiles, with the long loops comprising more than a single soliton each and the $\beta$-strands supporting a soliton that accounts for their bending. In the S1 File we present the parameter values of the free energy function (5) that supports the multi-soliton solution as its energy minimum. The Fig 2 Panels a)-c) summarizes the multi-soliton model we have constructed. Fig 2a) shows an overlay comparison between the C$\alpha$ trace of the PDB structure 2J6B and its multi-soliton model. The root-mean-square-distance (RMSD) between the two is 1,23 Å, comparable with the reported resolution 1.3 Å of the experimental structure. Fig 2b) shows a comparison of the C$\alpha$ trace bond angles (3) and torsion angles (4) between the PDB structure and its multi-soliton model. Fig 2c) identifies the secondary structures (according to PDB) and shows how we divide them into 20 individual solitons.

Fig 3 show our results for a particular heating and cooling simulation. (In the Fig 6b) we identify this simulation with an arrow, among all the displayed simulations).

The heating always starts from the multi-soliton that models 2J6B; the folding events we observe are fully reversible. The solid lines in the Fig 3 denote the evolution of the mean value over all structures in the simulated ensemble. The spread around each solid line shows the extent of one standard deviation around the mean. We note that in the high temperature stage we have random structures that reside in the phase of a self-avoiding random walk, with no regular structural details and in particular no structural resemblance to the folded conformation.

The lower horizontal axis in the Fig 3 gives the logarithmic value of the temperature factor $\beta$ during the simulation. The upper horizontal axis gives a corresponding Celsius value, that we deduce from a comparison with myoglobin simulations in [43]; the Celsius value should not be taken literally, it is intended to be suggestive, and for an accurate comparison between the simulation temperature and the corresponding Celsius value additional folding and unfolding experiments need to be performed over an extended range of temperature values. For a detailed analysis how to derive a relation between the temperature factor $\beta$ and the physiological temperature value measured in Celsius, we refer to [42].

The Fig 3a) and 3b) show temperature dependence of the C$\alpha$ root-mean-square distance (RMSD) between the 2J6B crystallographic structure and the multi-soliton, during the heating a) and cooling b) phases of the simulation. The final average RMSD value shown in the Fig 3b) at the end of the cooling has $\sim 1.3$ Å deviation from the initial multi-soliton, with $\sim 2.5$ Å one-sigma spread around the average. This deviation is comparable to the resolution in the experimental data; notably the final ensemble of our simulation includes all the final structures, including the small portion of configurations that do not support a slipknot. In Fig 3c) and 3d) we show the corresponding results for the radius of gyration value $R_g$.

In the Fig 3e) we show the evolution of $R_g$ at low temperatures, during the very early stages of the heating simulation. We observe a very slight but systematic initial decrease in the $R_g$ value. This means that when the chain starts unknotting, its effective volume initially shrinks, before the chain then expands and unfolds: The initial slipknotted multi-soliton structure appears to be (slightly) swelled in comparison to an unknotted but collapsed structure, even though the free energy of the slipknot has a lower value. As shown in Fig 3f) we observe a corresponding, systematic but very small increase of $R_g$ i.e. swelling at the end of the cooling period, before the energy minimum is reached.

Even though the swelling is very small in terms of the radius of gyration value $R_g$, at the visual level of the structure the effect is much more clear. The Fig 4 shows a structure at the minimal value of $R_g$ together with the fully folded conformation; the configuration with minimal $R_g$ value appears more compact.

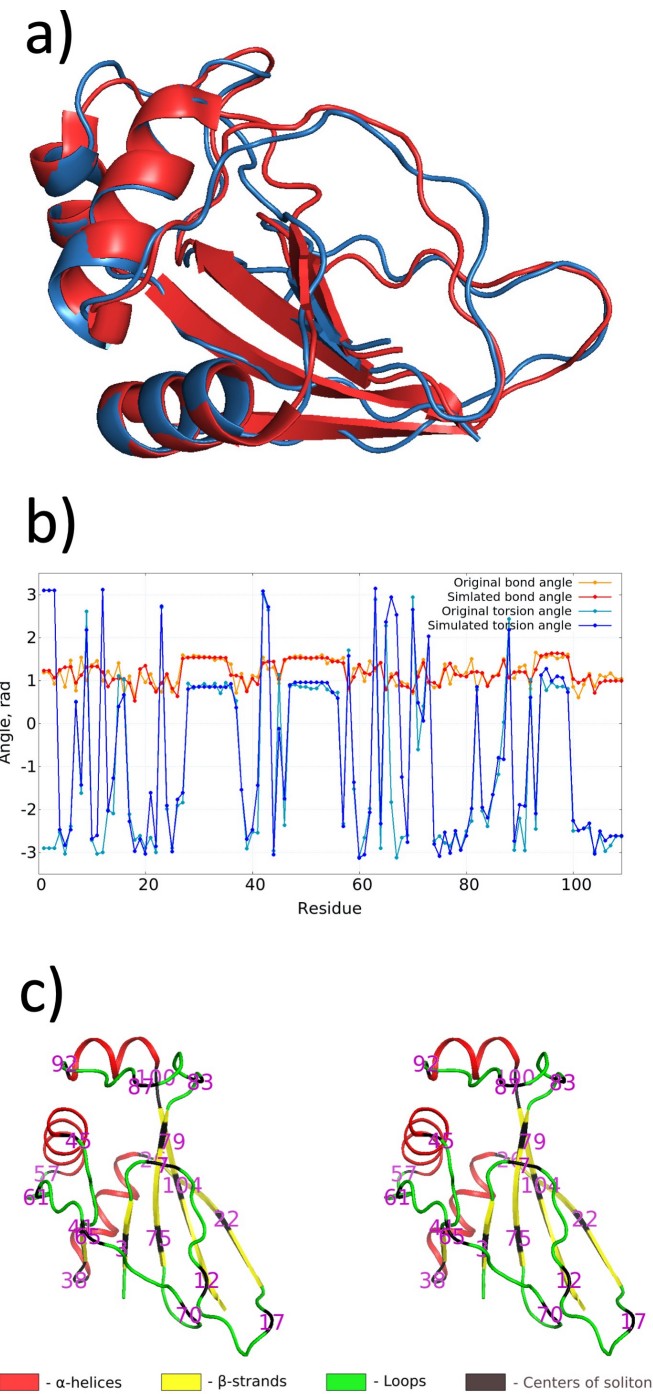

**Fig 2.** Panel a) shown overlay comparison of PDB structure 2J6B (red) and its multi-soliton structure (blue). The RMS distance between the two structures is 1.23 Å. Panel b) shows the virtual C$\alpha$ trace bond and torsion angles, for the PDB structure 2J6B and for its multi-soliton model. Panel c) is a cross-eye 3D view that identifies the secondary structures of 2J6B, together with the individual soliton centers.

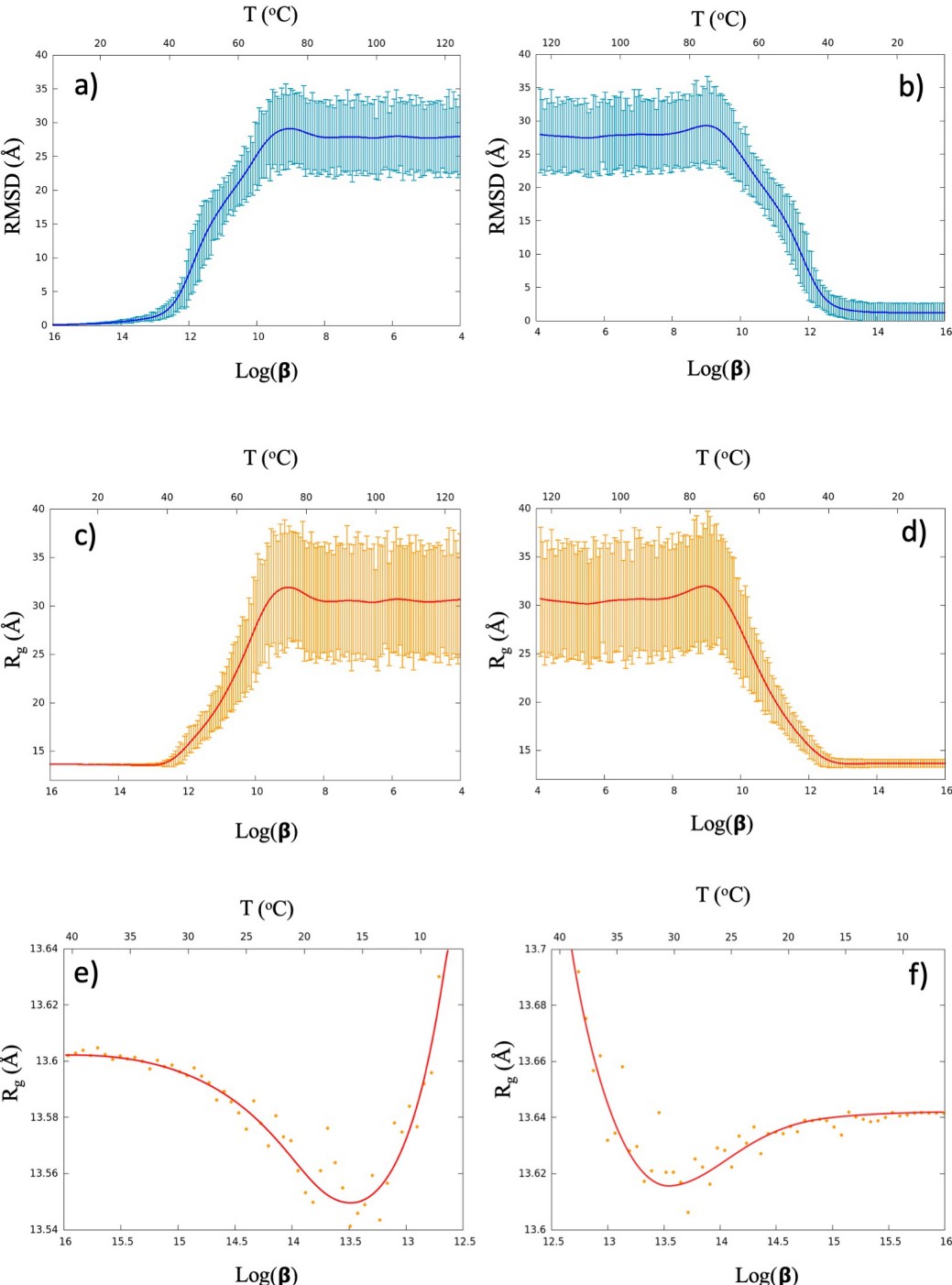

**Fig 3.** Panel a) shows the evolution of radius of gyration $R_g$ of the multi-soliton during the heating stage, and Panel b) shows the $R_g$ evolution during the cooling stage. The solid line shows the mean value of the simulation ensemble that consist of 1.000 independent cycles, and the spread denotes one standard deviation. Panel c) and d) show the corresponding results for the root-mean-square distance (RMSD) to the multi-soliton solution. Panels e) and f) are close looks to the $R_g$ values, at early/late stages of heating/cooling. There is a clear albeit very slight shrinking at the beginning of the unfolding process, and a similar swelling just before the final slipknot forms. The unfolding always starts from the multi-soliton. But for folding statistics the entire simulation ensemble is included, including those that do not form a slipknot.

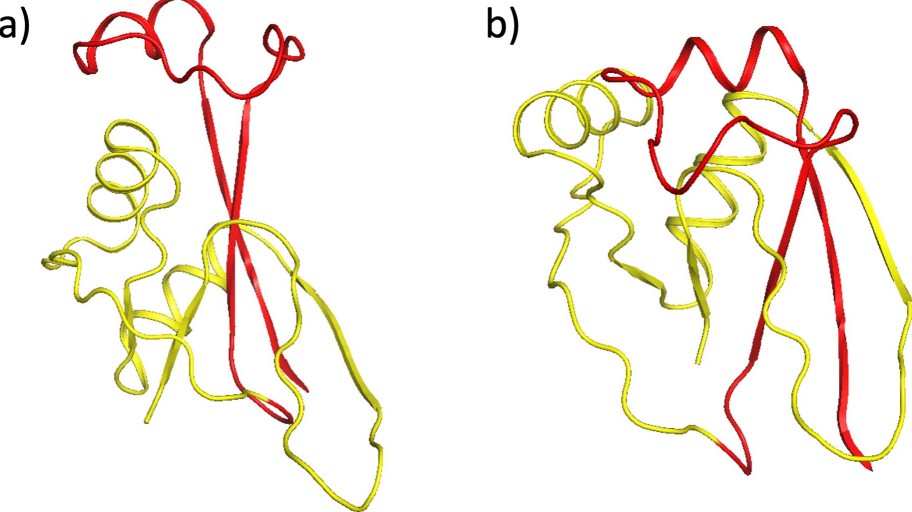

**Fig 4. The Panel a) shows the native folded conformation, and the Panel b) shows the conformation with the minimal $R_g$ value.**

A swelling that is caused by knottiness has been previously reported in an analysis of knotted crystallographic PDB structures [44].

We have analyzed the folding and unfolding transitions by following the temperature dependent fluctuations $\Delta\phi$ in the values of the torsion angle (4). In terms of $\Delta\phi$ the initial unfolding process appears to start at the location of the $\beta$-bridge that is centered at the proline with PDB residue number 61. In Fig 5a) we show how initially, at very low temperature values, the amplitude of the thermal fluctuations $\Delta\phi(61)$ slowly increases. This increase coincides with the decrease in the radius of gyration $R_g$ shown in Fig 3e) and at the same time we observe that the slipknot starts opening. When the temperature factor reaches a value $\log_{10}\beta \approx 13.0$ we observe a sharp order-disorder transition in the fluctuation amplitude $\Delta\phi(61)$. At this $\beta$-value the slipknot opens, and becomes converted into a trefoil knot shown in Fig 5b); see also S1 Movie. When the temperature factor further increases, the trefoil unknots and the chain starts unfolding, with a rapid, sharp increase in the radius of gyration value $R_g$ as shown in Fig 3e).

The folding transition proceeds similarly but in the opposite direction, the slipknot forms through an intermediate trefoil knot.

The Fig 6a) shows the simulated temperature dependence in $\Delta\phi$ along the entire chain. The first entire regular structure that melts is the $\beta$-strand that is located between PDB residues 74-80. This strand forms the edge of the looped segment that is threaded into a slipknot. In general, the $\beta$-strands appear to unfold at lower temperatures than the $\alpha$-helices, and the third $\alpha$-helix between residues 93-99 appears to be the last to unfold as temperature increases.

Previous authors have investigated the folding of knotted proteins, mostly using different variants of $C\alpha$ structure based Gō models. These authors do not report on trefoil knot folding intermediate, but they report significant difficulties in reaching the natively folded knotty conformation. Apparently, a knotted fold either does not appear at all [5] without the help of funneling or other kind of augmentation during the folding process, or then it occurs only very rarely, in about 1-2% of folding simulations [6] (2.8% in the case of AFV3-109 [6]). We have performed a series of folding and unfolding simulations under different conditions and Fig 6b) summarizes our results: When cooling proceeds very quickly the relative number of slipknots is small. However, when the cooling process becomes progressively slower, the number of slipknotted final conformations increases rapidly and as shown in the Fig 6b) we reach

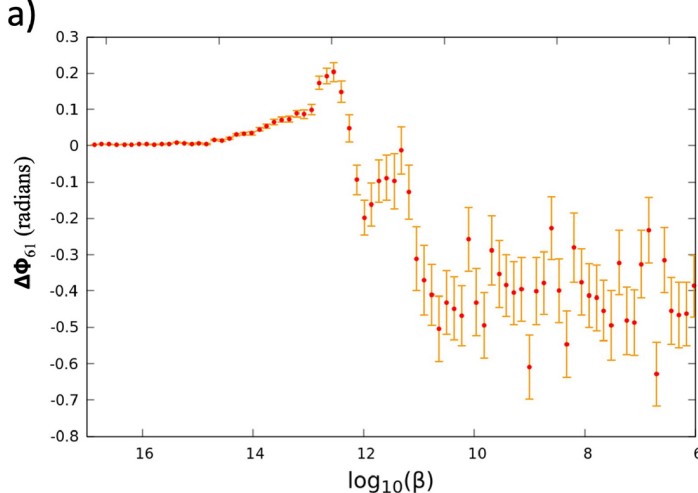

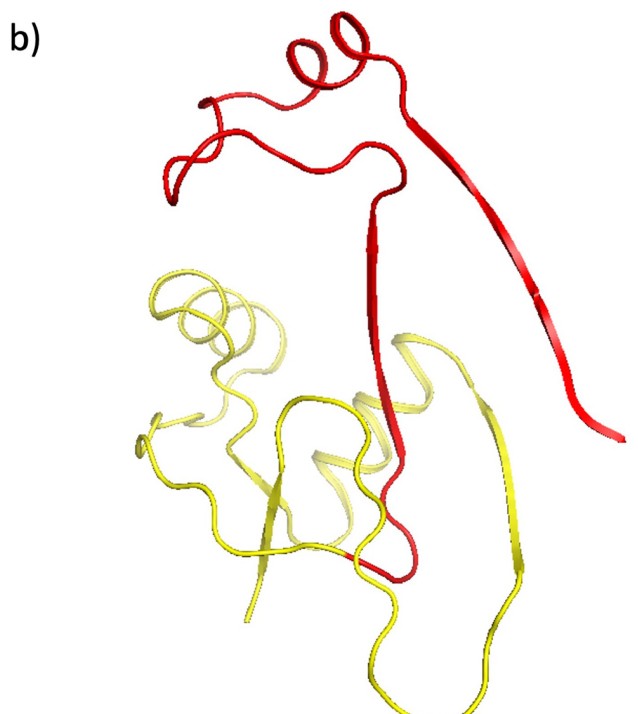

**Fig 5.** The Panel a) shows the initial increase in the thermal fluctuations $\Delta\phi_{61}$ of the torsion angle located at the proline with residue number 61 (the mean value of $\phi_{61}$ and its one standard deviation at the given $\log_{10}(\beta)$ value). The peak that is visible at $\log_{10}(\beta) \sim 13 - 12.5$ identifies the $\beta$-values where we observe an intermediate trefoil knot. The Panel b) shows the trefoil knot folding intermediate identified in the Panel a). The Panel shows that the trefoil knot forms, when the *C*-terminal of the looped segment pulls out from the knotting loop.

around 95 per cent success rate in the long and slow cooling simulations that we have performed. Thus we conclude that the slipknotted AFV3-109 is a slow folder [45, 46], proceeding through a trefoil knot folding intermediate. Moreover, since we observe both a trefoil knot folding intermediate and a light swelling at the final stages, the folding of AFV3-109 is not a pure relaxation (Arrhenius) process [47].

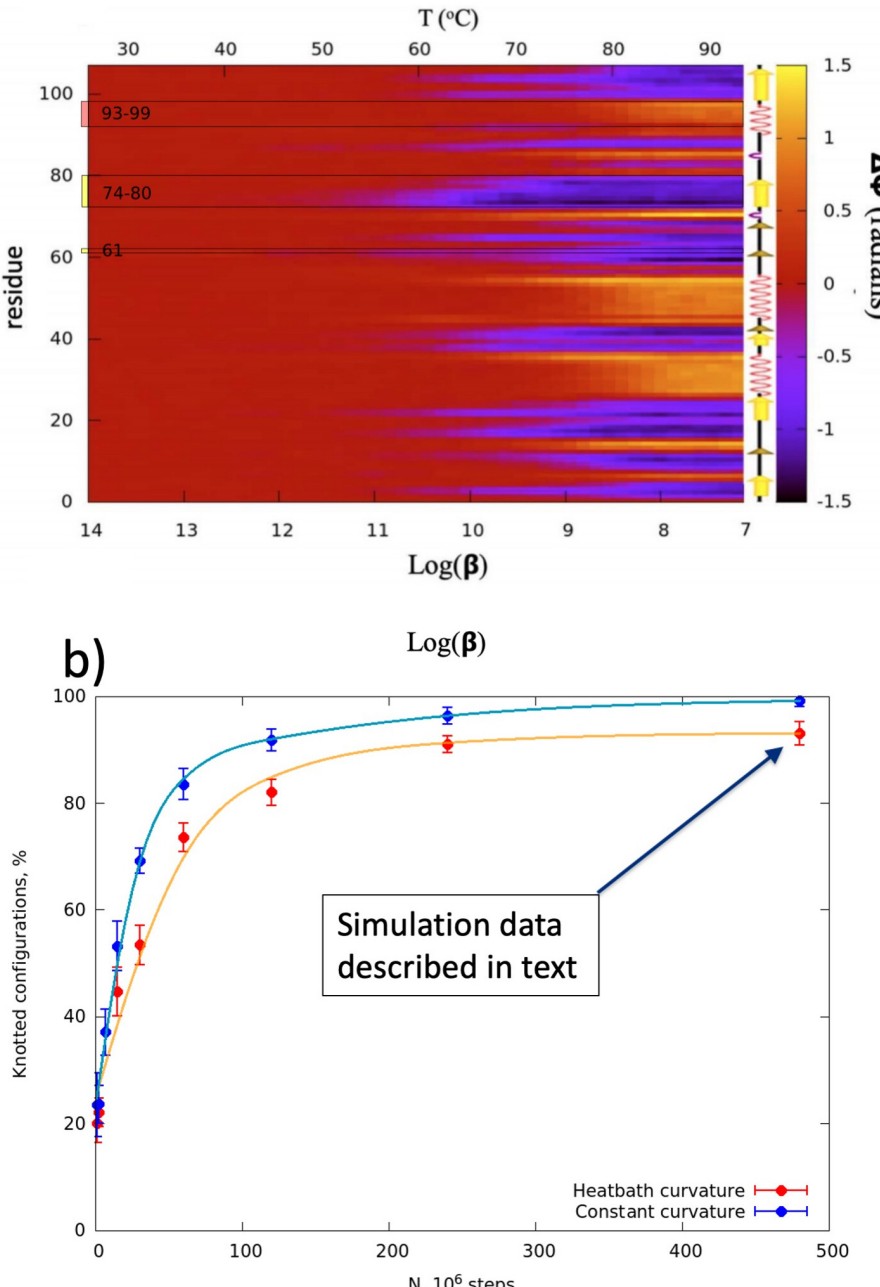

**Fig 6.** Panel a) characterizes the thermal fluctuations $\Delta\phi$ in the torsion angle values along the entire chain during the entire heating stage. The $\Delta\phi$ value is defined as the difference between the native value of the torsion angle $\phi$ and the ensemble average of the corresponding angle at the given $\log_{10}(\beta)$ value. The $\beta$-bridge with residue 61, together with the segments 74-80 and 93-99 discussed in the text, have been marked. Panel b) shows the relative number (in percentage) of slipknotted structures in the final folded ensemble, as a function of Monte Carlo steps after one full heating-cooling cycle. The arrow identifies the ensemble that we have analyzed in detail; in this ensemble around 94 per cent of final structures fold into the slipknotted multi-soliton of 2J6B. A movie that shows the (un)folding of a slipknotted structure can be found in S1 Movie. There we also show two examples of misfolding trajectories. In the S2 Movie with no knotted structure in the final stage, and in the file S3 Movie the misfolded state is a trefoil. Misfolding into an unknotted structure is a generic misfolding event. Despite appearing as a folding intermediate, a misfold into a trefoil final conformation is a *very* rare event in our simulations.

Finally, when we increase the relative stiffness of the bond angles, the rate of the slipknotted final conformations increases: We always observe both the trefoil folding intermediate and the slipknot final fold, at the end of our long and slow simulations in the limit where the bond angles have constant values during the entire heating and cooling process, when only torsion angles are truly mobile. In actual proteins the virtual C$\alpha$ backbone bond angles are known to be very stiff and the torsion angles are known to be quite flexible. Our simulation results show that when one properly accounts for the relative stiffness/flexibility of the bond and torsion angles, the C$\alpha$ backbone of AFV3-109 practically always folds into a slipknot; misfolded conformations are indeed very rare.

## Conclusions

Topology and in particular self-entanglement play an important role in protein folding and dynamics. But topological effects are difficult to investigate. Moreover, conventional simulation approaches aim to describe a protein and its folding as a local process, at atomic level precision. Due to limitations in available computational resources it then becomes very difficult to detect large scale collective motions and global, topological phenomena in the conventional simulation approaches. Nevertheless, knotty and other kind of self-entanglements are often important to protein stability, and presumably also important for the correct biological function. Thus there is value to develop global, topological approaches to protein folding and dynamics, as a complement to local, atomic level scrutiny based approaches.

We have developed a global technique that is rooted on topological concepts, to analyze and describe the formation of topological structures in proteins, in particular aspects of knottiness and self-entanglement. For this, we have modeled the entire C$\alpha$ backbone of a protein in terms of a single topological multi-soliton entity; the multi-soliton describes the minimum of a mechanical free energy. As a case study, we have investigated the folding and unfolding of the slipknotted AFV3-109 protein, instigated by variable ambient temperature, using powerful state-of-art Monte Carlo techniques of non-equilibrium thermodynamics. We have found that the multi-soliton describes the formation of the slipknot very accurately, and we are able to describe the folding pathways and make predictions on the physical origin of knot formation. In paricular, we have been able to observe a trefoil knot as a folding intermediate. Our results demonstrate the value of developing global approaches to protein folding and dynamics; global approaches are highly accurate, and even though they may lack in atomic level details they appear to correctly capture the global, topological aspects of self-entanglement during protein folding and dynamics.

## Supporting information

**S1 File. Theoretical/Technical background and data file.**
(PDF)

**S1 Movie. Unfolding of slipknotted structure, with increasing temperature, as described in the text.**
(MP4)

**S2 Movie. Misfolding into an unknotted structure; this is generic misfold, see Fig 6.**
(MP4)

**S3 Movie. Misfolding into a trefoil structure; this is a very rare misfold event, see Fig 6.**
(MP4)

## Author Contributions

**Conceptualization:** Alexander Molochkov, Antti J. Niemi.

**Data curation:** Alexander Begun, Sergei Liubimov.

**Formal analysis:** Alexander Molochkov, Antti J. Niemi.

**Investigation:** Alexander Begun, Alexander Molochkov, Antti J. Niemi.

**Methodology:** Antti J. Niemi.

**Project administration:** Alexander Molochkov, Antti J. Niemi.

**Resources:** Alexander Molochkov.

**Software:** Alexander Begun, Sergei Liubimov, Alexander Molochkov.

**Supervision:** Alexander Molochkov, Antti J. Niemi.

**Visualization:** Alexander Begun, Sergei Liubimov, Alexander Molochkov.

**Writing – original draft:** Antti J. Niemi.

**Writing – review & editing:** Antti J. Niemi.

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
